# The Efficacy of Convalescent Plasma Use in Critically Ill COVID-19 Patients

**DOI:** 10.3390/medicina57030257

**Published:** 2021-03-11

**Authors:** Livius Tirnea, Felix Bratosin, Iulia Vidican, Bianca Cerbu, Mirela Turaiche, Madalina Timircan, Madalin-Marius Margan, Iosif Marincu

**Affiliations:** 1Department of Infectious Diseases, “Victor Babes” University of Medicine and Pharmacy, 300041 Timisoara, Romania; liviustirnea@yahoo.com (L.T.); iulia.georgianabogdan@gmail.com (I.V.); ionitabiancaelena@yahoo.com (B.C.); mirela.turaiche@gmail.com (M.T.); imarincu@umft.ro (I.M.); 2Department of Gynecology, “Victor Babes” University of Medicine and Pharmacy, 300041 Timisoara, Romania; timircan.madalina@yahoo.com; 3“Victor Babes” University of Medicine and Pharmacy, 300041 Timisoara, Romania; marganmm@gmail.com

**Keywords:** convalescent plasma, COVID-19, critically ill, transfusion, SARS CoV-2

## Abstract

*Background and Objectives:* On 24 March 2020, the United States Food and Drug Administration (FDA) announced the approval of convalescent plasma therapy for critically ill patients with severe acute respiratory syndrome coronavirus 2 (SARS-CoV-2) as an emergency investigational new drug. This pilot study from Romania aimed to determine if convalescent plasma transfusion can be beneficial in the treatment of selected critically ill patients diagnosed with a SARS-CoV-2 infection. *Materials and Methods:* Donor and receiver eligibility for critically ill coronavirus disease 2019 (COVID-19) patients was based on Romanian guidelines issued at the time of the study. Here, we describe the evolution of a total of five eligible patients diagnosed with COVID-19 who received convalescent plasma (CP) in Romania. *Results:* In spite of our efforts and convalescent plasma administration, three of the five patients did not survive, while the other two recovered completely. Over the course of our five-day laboratory record, the surviving patients had significantly lower values for C-reactive protein, interleukin-6, and white blood cells. *Conclusions:* This pilot study provides insufficient evidence to determine the efficacy of convalescent plasma use as a therapeutic option for critically ill COVID-19 patients.

## 1. Introduction

The coronavirus disease 2019 (COVID-19), a disease caused by the severe acute respiratory syndrome coronavirus 2 (SARS-CoV-2), has turned into a rapidly evolving pandemic [1]. On 11 March 2020, the World Health Organization (WHO) declared the novel coronavirus disease (COVID-19) a global pandemic. At the time of writing, 28 September 2020, there are no approved specific and efficient treatments SARS-CoV-2 infection. More than 1,000,000 deaths have been recorded worldwide, and the number of confirmed patients and fatalities has been rising. Treatments with antivirals like Remdesivir^®^, Favipiravir^®^, or Tocilizumab^®^ (a monoclonal antibody directed against the interleukin-6 receptor), have been proposed and are still under investigation.

From the 1880s to the antibiotic era, convalescent blood products (CBP) have been used to prevent and treat many bacterial and viral infections in humans and animal models [2]. The potential of convalescent plasma (CP) was described during a wide range of viral infections, including measles, parvovirus B19, Ebola, the 2019 influenza A virus subtype H1N1 pandemic, and the Middle East Respiratory Syndrome Coronavirus (MERS-CoV). CP is obtained by collecting blood from patients who recovered from illness and developed antibodies. The mechanism involves passive antibody administration to prevent or treat infectious diseases.

With no proven efficient treatments, CP has been used as adjuvant therapy for patients diagnosed with COVID-19 in selected cases worldwide [3]. Due to various barriers, CP is not widely used in Romania. Considering the pool of potential recipients in our region, chosen on the basis of the Romanian guidelines for convalescent plasma administration to critically-ill COVID-19 patients and the availability of donors, we designed and implemented a pilot study to observe the initial clinical experience with CP and determine its potential use in eligible patients who were diagnosed with a severe form of COVID-19.

## 2. Materials and Methods

This pilot study for CP use in Romania was conducted at the Infectious Disease Department, Dr. Victor Babes Clinical Hospital for Infectious Diseases and Pulmonology in Timisoara, from May 2020 to September 2020. The local ethics committees approved the study, and each patient gave written informed consent.

### 2.1. Patients Enrolment

According to the methodology elaborated by the Health Minister of Romania, published on 22 April 2020, for the collection, testing, processing, storage, and distribution of plasma from the cured donor of COVID-19 and its monitored use for critical patients with COVID-19 [4], patients are eligible to receive convalescent plasma treatment if they meet the following criteria: (1) critically ill patients confirmed with COVID-19 by the RT-PCR method for which an obtained consent form was signed either by the patient or by the legal person who agreed to the administration of freshly frozen plasma from a donor cured of COVID-19 (PPC-DV-COVID-19); (2) being at least 18 years of age; (3) being admitted to the Intensive Care Unit for a rapid progression of severe pneumonia with pulmonary infiltrates covering over 50% of the lung fields in 24–48 h despite the administered maximal treatment; (4) being mechanically ventilated for less than 10 days or on the threshold of intubation and mechanical ventilation due to having severe dyspnea, increased respiratory rate, oxygen saturation ≤93% with oxygen face mask, paCO2 > 55 mmHg in patients without Chronic Obstructive Pulmonary Disease (COPD). Plasma collection and handling were done by experienced staff at Timisoara Regional Transfusion Center using standard operating procedures in plasma collection according to national guidelines. The plasma was labeled as: “Freshly frozen plasma from the donor cured of COVID-19 (PPC-DV-COVID-19)” and on the final label of the unit: “To be used exclusively for patients with COVID-19”. The patients also received antiviral agents, broad-spectrum antibiotics, and steroids continuously until the SARS-CoV-2 viral loads became negative or the patients deceased. A total of 29 critically ill patients were enrolled in the study based on the consent form, out of which only 5 male patients, ranging from 40 to 46 years of age, with severe COVID-19 were eligible according to all the above criteria. Convalescent plasma was administered between 4 and 10 days after admission.

### 2.2. Donor Eligibility

Potential plasma donors were selected from individuals able to prove their history of SARS-CoV-2 infection by: (1) a discharge note or medical letter mentioning the diagnosis of COVID-19 confirmed by RT-PCR and a confirmation of recovery by a negative RT-PCR test (nasopharyngeal swabs), who could donate CP at least 14 days after the confirmation of recovery, (2) a discharge note or medical letter in which the diagnosis of COVID-19 confirmed by RT-PCR was mentioned and in which the isolation period subsequent to the discharge on clinical and biological criteria was recorded, who could donate CP 14 days from the end of the isolation period; (3) a positive RT-PCR test for SARS-CoV-2 infection and a document issued by the national public health directorates attesting the end of the isolation period, in the conditions in which the donor was not hospitalized, who could donate 14 days from the end of the isolation period; (4) a positive RT-PCR test for SARS-CoV-2 infection performed at least 28 days earlier in conditions in which the patient was not hospitalized or registered in the national and public health departments; (5) a positive result in a serological test of anti-SARS-CoV-2 IgG antibodies, performed voluntarily, in the case of asymptomatic persons whose infection had not been confirmed by a positive RT-PCR test, who could donate at least 14 days from the date of the test; (6) the absence of blood-borne infections and anti-HLA antibodies that can increase the risk of transfusion-related acute lung injury (TRALI). It was mandatory to sign the informed consent to enter the selection procedure for donation of plasma by plasmapheresis and of whole blood with the coded transmission of the data on the donation in the national and European database.

### 2.3. Disease Severity Classification

The following criteria were applied to define severely ill patients: (1) respiratory failure requiring mechanical ventilation, (2) respiratory rate > 30/min, (3) Saturation of Hemoglobin with Oxygen (SpO_2_) < 90%, (4) Arterial Oxygen Partial Pressure/Fractional Inspired Oxygen (PAO_2_/FIO_2_) < 300 (PAO_2_ measured in mm Hg, and FIO_2_ measured as the fraction of inspired oxygen), (5) severe lymphocytopenia, (6) elevated lactate levels (>2 mmol/L), (7) inflammatory syndrome defined by the following: C-reactive protein (CRP) > 100 mg/dL, elevated ferritin and elevated interleukin 6 levels, (8) coagulation disorders including high D-dimer levels, platelet count < 100,000/µL, fibrinogen < 2 mg%, (9) failure of other organs requiring admission to the intensive care unit (ICU), (10) ground-glass opacities involving > 50% of the lungs at chest X-ray. 

### 2.4. Statistical Analysis

Data were analyzed using IBM SPSS version 26.0.0 for Windows. The Student’s *t*-test for two samples was used to compare means for continuous variables analysis. Here, we compared the average lab values of the two patients who survived and of the other three who died, including CRP, procalcitonin, and IL-6 levels and white blood cells and lymphocyte counts. The laboratory analyses for these five variables were performed one day before CP transfusion and daily for the next five days after the transfusion.

## 3. Results

### 3.1. Clinical Characteristics

The most common symptoms during hospitalization were cough (4/5 patients), shortness of breath (3/5), and fever (4/5). None of the patients was a smoker, and two patients had coexisting chronic diseases at admission, including hypertension and atrial fibrillation. Table 1 lists the clinical characteristic of the patients, including the treatments received before and after CP transfusion. All five patients were treated with various antiviral agents and other supportive care (Table 1). The most commonly used antivirals were Remdesivir^®^ (3/5); one patient received Kaletra^®^, and another Darunavir^®^ with Norvir^®^. Antibiotics and corticosteroids are part of our national protocol, so they were administered to the patients since their admission to the hospital. Chest computed tomography (CT) scans were performed on all patients and demonstrated that all five patients presented bilateral multiple ground-glass opacities.

### 3.2. Laboratory Findings

The SARS CoV-2 viral load, estimated by the cycle threshold (Ct) value from reverse transcription-polymerase chain reaction (RT-PCR), was positive in five patients before CP transfusion and decreased to negative in patient number one after seven days and in patient number three after 12 days. The levels of C-reactive protein and procalcitonin, two inflammatory biomarkers, were already increased before plasma transfusion. After the transfusion, we observed a declining trend following CP treatment in two out of five patients (Table 2). The levels of interleukin-6 (IL-6), a pro-inflammatory cytokine, increased in all five patients and decreased in two out of five patients after plasma transfusion (Table 3). Lymphocytopenia was common to all five patients, and we observed an increase in lymphocyte count after five days. White blood cell count was increased in all five patients. The values colored in red represent the abnormal findings and changes.

### 3.3. The Outcome of Patients Treated with CP

All five patients were critically ill with a severe form of COVID-19 disease. Two patients were discharged from the hospital: patient number one after 15 days, and patient number three after 25 days. The other three patients died. The Acute Physiology and Chronic Health Evaluation II score was calculated for all patients, and the plasma doses consisted of a 400 mL pack for all patients.

The APACHE II (Acute Physiology and Chronic Health Evaluation II) score was calculated within 24 h after admission to the ICU for all our patients. The score is calculated using 12 parameters: PaO2 (depending on FiO_2_), temperature, heart rate, respiratory rate, sodium, serum potassium levels, creatinine value, hematocrit value, white blood cell count, and Glasgow Coma Scale score. A higher score corresponds to a more severe disease and a higher risk of death, and all three patients that subsequently did not survive had their APACHE II scores higher than 15.

Significant mean differences between the survivors and the deceased (Table 4) were identified for CRP, procalcitonin, and white blood cells count. After the transfusion, C-reactive protein levels in patients who did not survive were maintained at values averaging 37 times higher than those for the patients who survived (CI: (−109.57; −25.66), *p* < 0.000). Other statistically significant differences between the survivors and the deceased were observed in IL-6 levels (CI: (−450.50; −185.33), *p* < 0.000), averaging 19 times higher values in the non-survivors, and white blood cell count (CI: (−12.43; −5.07), *p* < 0.000), averaging 3 times higher values in the non-survivors (Figure 1).

## 4. Discussion

Previous studies have suggested that CP could serve as a treatment for various viral infections, including influenza A H1N1 [5], the severe acute respiratory syndrome (SARS) of 2003 [6], avian influenza A [7], and other viral infections. Furthermore, in 2015, a study conducted by Mair-Jenkins et al. [8] concluded that patients with SARS who were treated with convalescent plasma had a significantly higher discharge rate and a lower mortality than the patients receiving steroid treatment. Despite a favorable historical record, few clinical trials with CP worldwide and none in our country have been performed, mainly because the use of convalescent plasma is indicated as an emergency treatment. Another limitation involves the fact that CP treatment is dependent on the number of potential seroreactive donors with high antibody titers. We cannot control the number of patients who will donate their plasma, the quantity of antibodies contained in the collected plasma, and the unpredictable way the disease evolves in a patient.

The relationship between antibody titers and their effectiveness in COVID-19 patients is unclear. A recent finding suggests that most donors who recovered from mild to moderate disease forms had low levels of SARS-CoV-2 antibodies and a modest neutralizing activity [9]. This is also confirmed by the results obtained in another study conducted by Hao Zeng et al., where the authors found that patients treated with CP with high neutralizing antibodies titer (NAT50 > 1:1640) showed a more obvious improvement than patients receiving CP with low NAT50 [10]. However, the connection between neutralizing anti-SARS-CoV-2 antibodies and total anti-SARS-CoV-2 antibodies is still a hypothesis that needs to be studied.

Five patients were included in our study, and three patients died. We believe that treatment timing and the degree of severity of the disease are important factors associated with mortality. Since the plasma donation protocol was published late in Romania (April 2020), plasma collection was delayed, and patients admitted to the ICU could not benefit from it or received it very late, when their disease was categorized as critical or end-stage. In our case report, two patients who received CP more promptly than the other three survived. A recent study published by Luchsinger et al. [11] concluded that CP therapy could not reduce mortality in critically ill patients with end-stage disease. With the same results, a randomized clinical trial with 280 patients [12] just terminated at the end of November, concluding that there were no statistically significant differences in the clinical status of the patients who received CP, in comparison to patients treated with placebo. Also, the mortality rates did not significantly differ, with 10.96% overall mortality for the patients who received CP, compared to 11.43% for the placebo group. Similar results were described by a Chinese trial [13] in a study involving severe cases of SARS-CoV-2 infections, which showed clinical improvement in 51.9% of CP-treated cases and in 43.1% of patients undergoing regular care, which was not a substantive change (*p*-value = 0.26) and confirms our findings as well.

On the contrary, an earlier randomized trial [14] on CP administration with a high titer of antibody to patients diagnosed with severe COVID-19 indicated that for the intervention group including 21 patients, the time to clinical improvement in a 28-day period was 4.94-day shorter than in the control group comprising 15 patients. The same positive results were observed in a multicentric interventional trial [15] assessing the efficacy of CP use in 46 patients suffering from severe COVID-19, which showed a statistically significant decrease in mortality rate in the CP group (6.5%) compared to the control group (15%).

Other clinical trials assessing CP use for COVID-19 patients were different from our study, since either they were treating moderate cases or they were halted due to various reasons. For example, the largest clinical trial to date that took place in India [16] on 464 patients did not show statistically significant results between CP use and conventional treatment, although the cases involved were of moderate severity. The ConCOVID trial [17] from the Netherlands prematurely stopped with insignificant differences in the two study arms regarding mortality, severity, or time spent in the hospital.

Another tool that we used to predict mortality was the APACHE II score, which is calculated within 24 h after admission to the ICU. We observed that some patients deteriorated rapidly, suffered multiple organ failure, and presented acute respiratory distress syndrome. In a study that included 178 patients with severely COVID-19 disease, Zou et al. [18] demonstrated the APACHE II score to be independently associated with hospital mortality. Furthermore, the APACHE II score predicted better hospital mortality than the SOFA and CURB65 scores. An APACHE II score greater than or equal to 17 serves as an early warning indicator of death, consistent with our study. Patient 1 and patient 3, who presented a score lower than 17, survived. In contrast, patients 2, 4, and 5 had a score equal to or greater than 17 and eventually died. This may help practitioners make clinical decisions, and our study recommend administering CP rapidly for better patient outcomes, but further research is needed. One limitation consists in the Glasgow Coma Scale as being an important criterion in APACHE II score calculation, but nerve damage was rarely reported in the study to properly assess the score.

Regarding our biological findings, we observed high levels of inflammatory biomarkers, including C-reactive protein and procalcitonin. Also, all five patients had high interleukin-6 levels, which is consistent with the findings of Evan et al. [19], who found that interleukin-6 is a stable indicator of poor outcome in patients with pneumonia and Acute Respiratory Distress Syndrome (ARDS). Lymphopenia has also been proven to be associated with disease severity in patients with COVID-19. All patients included in our study had a low lymphocyte level. The mechanism leading to this involves cell destruction, massive lymphocyte recruitment in the infected airways, and cytokine-mediated cell death [20].

Other limitations of our study should be noted. First, this study was constrained by the small sample size, and CP-treated patients were not compared with a control group of patients. Second, the patients received concomitant therapies, including antiviral agents, antibiotics, and corticosteroids, making it impossible to discriminate the specific contribution of CP to the clinical course or outcomes. Convalescent plasma was administered 2 to 9 days after admission to the hospital. The relationship between transfusion time and clinical outcomes should be further investigated. Moreover, patients included in the study received different CP doses, and we did not know the antibody titers of the received plasma due to a lack of the required resources.

The most difficult challenge involved finding donors. Although each discharged and eligible patient files a form in our hospital in which we require acceptance to transmit personal data (e.g., name, telephone number, the first positive RT-PCR result) to the local transfusion center, we face a growing need for donors. Unfortunately, many rules have to be applied in the social distancing era before one can donate plasma, including checking the temperature before entering the building, maintaining social distancing, screening donors on the telephone, having donors leave work for a day and meet several requirements.

## 5. Conclusions

The current study provides insufficient evidence to prove or disprove the efficacy of convalescent plasma use as a therapeutic option for critically ill COVID-19 patients. Because of the small number of patients involved, the concomitant treatment modalities, and other limitations, there is no clear evidence that convalescent plasma improves patients’ outcomes.

## Figures and Tables

**Figure 1 medicina-57-00257-f001:**
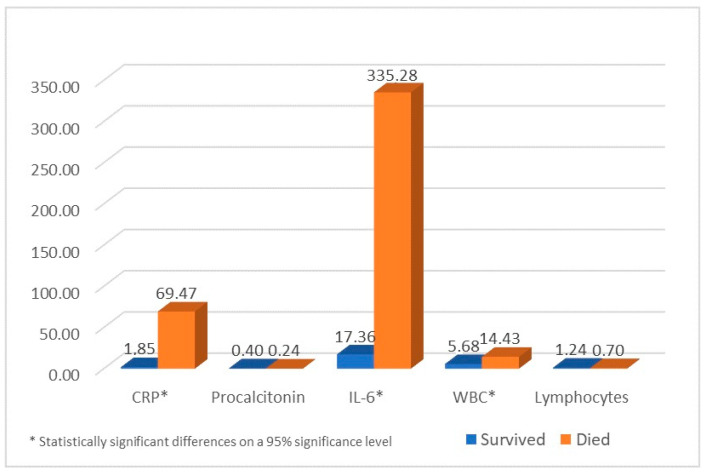
Average values of laboratory findings in survivors and deceased patients.

**Table 1 medicina-57-00257-t001:** Clinical characteristics and associated treatment of patients who received convalescent plasma (CP).

Patient	1	2	3	4	5
***Sex***	Male	Male	Male	Male	Male
***Age (years)***	45	46	40	46	43
***Weight (kg/BMI)***	80/21.4	95/27.6	100/32.3	103/35.1	87/24.4
***Comorbidities***	No	hypertension	obesity	obesity, atrial fibrillation	No
***Time until admission***	7 days	5 days	6 days	6 days	8 days
***Symptoms***	cough, shortness of breath	fever, cough, shortness of breath	cough, sputum production, fever	confusion, cough, fever	fever, fatigue, shortness of breath
**Treatment**					
***Antivirals***	Kaletra	Remdesivir	Darunavir, Norvir	Remdesivir	Remdesivir
***Corticosteroids***	Dexamethasone	Dexamethasone	Dexamethasone	Dexamethasone	Dexamethasone
***Antibiotics***	Cefort, Moxifloxacin	Cefort	Moxifloxacin	Moxifloxacin	Moxifloxacin
***Others***	Fraxiparine, Tocilizumab	Fraxiparine	Fraxiparine	Fraxiparine	Fraxiparine

**Table 2 medicina-57-00257-t002:** Laboratory results before and after CP transfusion.

Patient	1	2	3	4	5
***Laboratory findings***					
***CRP level mg/dl (normal levels < 5)***					
*Before CP transfusion*	2.19	172.2	5.36	176.5	120.56
*Day 1 posttransfusion*	0.87	37.11	4.59	251.9	40.3
*Day 2 posttransfusion*	0.53	16.55	2.65	106.2	66.44
*Day 3 posttransfusion*	0.31	19.75	3.12	34.11	41.71
*Day 4 posttransfusion*	0.33	28.3	1.27	9.51	16.16
*Day 5 posttransfusion*	0.57	94.99	0.35	7.23	10.90
***Procalcitonin ng/mL (normal range 0–0.5)***					
*Before CP transfusion*	1.2	0.08	0.05	0.06	0.71
*Day 1 posttransfusion*	0.9	0.22	0.04	0.06	0.09
*Day 2 posttransfusion*	1.1	0.20	0.23	0.04	0.11
*Day 3 posttransfusion*	0.7	0.23	0.04	0.04	0.09
*Day 4 posttransfusion*	0.3	0.54	0.03	0.10	0.06
*Day 5 posttransfusion*	0.2	0.70	0.02	0.87	0.14
***IL-6 pg/mL (normal range 0–7)***					
*Before CP transfusion*	63.1	168.3	2.77	251.9	189.3
*Day 1 posttransfusion*	51.7	350.9	3.11	152.3	205.7
*Day 2 posttransfusion*	32.8	169.9	2.15	78.2	212.24
*Day 3 posttransfusion*	25.8	327.5	2.10	458	360.79
*Day 4 posttransfusion*	15.2	483.2	1.68	443.9	333.60
*Day 5 posttransfusion*	6.7	1089	1.20	343.6	416.70
***WBC count × 10^9^/L (normal range, 4–10)***					
*Before CP transfusion*	7.47	24.06	5.50	6.55	7.2
*Day 1 posttransfusion*	6.77	13.53	5.25	7.36	13.65
*Day 2 posttransfusion*	6.31	9.58	4.56	12.00	12.12
*Day 3 posttransfusion*	6.30	11.35	3.58	13.69	16.20
*Day 4 posttransfusion*	7.92	22.23	3.89	13.68	21.65
*Day 5 posttransfusion*	7.45	26.10	3.12	8.67	20.20
***LY count × 10^9^/L (normal range, 1.1–3.2)***					
*Before CP transfusion*	0.65	0.48	0.73	0.74	0.12
*Day 1 posttransfusion*	0.72	0.88	0.68	1.25	0.19
*Day 2 posttransfusion*	1.44	0.62	0.49	0.62	0.28
*Day 3 posttransfusion*	2.30	2.5	0.34	0.11	0.25
*Day 4 posttransfusion*	4.33	0.78	0.25	0.75	0.64
*Day 5 posttransfusion*	2.55	0.79	0.36	0.98	0.66

CRP, C-reactive protein; IL-6, interleukin-6; WBC, white blood cell; LY, lymphocyte.

**Table 3 medicina-57-00257-t003:** Characteristics and outcome of patients treated with CP.

Patient	1	2	3	4	5
*Complications before CP administration*	hepatic cytolysis, metabolic acidosis, hypokalemia, basal pleurisy	sepsis of unknown cause, hyponatremia	anemia, left pleurisy	sepsis of unknown cause, hepatic cytolysis	anemia
*APACHE II score*	3	17	10	20	19
*Clinical classification before CP transfusion*	severe	severe	severe	severe	severe
*Transfusion volume*	400 mL	400 mL	400 mL	400 mL	400 mL
*The interval between admission and plasma transfusion*	2	7	3	8	9
**Mechanical Ventilation**					
*Intubated, days before CP*	No	3	No	5	no
*Extubated, days after CP*	NA	NA	NA	NA	NA
*Clinical outcome*	Survived	Died	Survived	Died	Died
*Length of hospital stay (days)*	15	21	25	11	11
*SARS CoV-2 viral load (cycle threshold, decreased to negative in days)*	7	NA	12	NA	NA

APACHE II, Acute Physiology and Chronic Health Evaluation II, SARS CoV-2, severe acute respiratory syndrome coronavirus 2.

**Table 4 medicina-57-00257-t004:** Statistical analysis comparing the surviving patients and the deceased.

Blood Test	Outcome	Mean	CI (95%)	*p*-Value
***CRP ****	Survived	1.84	[−109.57; −25.66]	<0.000
	Died	69.46		
***Procalcitonin***	Survived	0.44	[−0.14; 0.46]	0.283
	Died	0.24		
***IL-6 ****	Survived	17.35	[−450.50; −185.33]	<0.000
	Died	335.27		
***WBC ****	Survived	5.67	[−12.43; −5.07]	<0.000
	Died	14.42		
***Lymphocyte Count***	Survived	1.12	[−0.13; 1.20]	0.115
	Died	0.70		

* Statistically significant differences on a 95% significance level.

## Data Availability

Data is available at request from the authors.

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
