# Peer review of "The Efficacy of Convalescent Plasma Use in Critically Ill COVID-19 Patients"

_medicina, 2021, doi:10.3390/medicina57030257_

Round 1
Reviewer 1 Report
Executive Summary
The revised manuscript titled “The efficacy of convalescent plasma uses in critically ill COVID-19 patients” describes clinical trials using convalescent plasma in the treatment of Covid-19 in Romania. Even though this research has a very limited number of patients (N=5) and the treatment does not show significant improvement in survival rate, I still believe it is a successful pilot study during the Covid-19 pandemic. Authors may perform minor revisions with the following details to further improve the reader friendliness of this manuscript.
Major Comments
There is no major comment or critique for this manuscript.
Minor Comments
- In Table 2, please highlight normal or abnormal levels with contrast color to improve the readers’ friendliness.
- In Table 3, Table 4, and Figure 1, please use consistent descriptions for “discharged and deceased” or “survived and died”.
- In Figure 1, please add asterisks for significantly different comparison groups based on P-value showed in Table 4.
Author Response
Dear reviewer,
Thank you for the feedback on the behalf of our team!
Here are the following changes made based on your advice:
- In Table 2 we highlighted the abnormal levels with contrasting dark red color.
- In Table 3, Table 4, and Figure 1, we have changed all descriptions to “survived and died”.
- In Figure 1, we have added asterisks for significantly different comparison groups based on P-value showed in Table 4. Also added a row in Table 4 to indicate the meaning of the asterisks.
Reviewer 2 Report
The authors have addressed the comments. If the manuscript is going to be published as a full paper and not as a letter to the editor/short report, then a more detailed discussion with more data on studies on convalescent plasma, timing of administration, choice of donors, quality of antibody response etc would be valuable.
Author Response
Dear reviewer,
Thank you for taking the time to analyse our article paper, and we greatly appreciate the useful feedback that allowed us to improve our paper. Thus, kindly review the following changes:
- In Table 2 we highlighted the abnormal levels with contrasting dark red color.
- In Table 3, Table 4, and Figure 1, we have changed all descriptions to “survived and died”.
- In Figure 1, we have added asterisks for significantly different comparison groups based on P-value showed in Table 4. Also added a row in Table 4 to indicate the meaning of the asterisks.
- Table 4 was mentioned on line 152
- As for publishing a full paper, we have added a more detailed discussion with more data on studies on convalescent plasma, administration timing, choice of donors, and quality of antibody response.
- The discussion section was extended from 904 words to 1163 words, including five more references adding more data on convalescent plasma use in clinical trials, discussing results, the timing of administration, the quantity of antibody titers, and the time of hospital stay.
- The number of references in our article paper increased from 15 to 20.
Best regards,
Dr. Felix Bratosin